# Towards Probing Conformational States of Y2 Receptor Using Hyperpolarized ^129^Xe NMR

**DOI:** 10.3390/molecules28031424

**Published:** 2023-02-02

**Authors:** Peter Schmidt, Alexander Vogel, Benedikt Schwarze, Florian Seufert, Kai Licha, Virginia Wycisk, Wolfgang Kilian, Peter W. Hildebrand, Lorenz Mitschang

**Affiliations:** 1Institute of Medical Physics and Biophysics, Medical Faculty, University of Leipzig, Haertelstrasse 16-18, 04107 Leipzig, Germany; 2Institute of Chemistry and Biochemistry, Freie Universitaet Berlin, Takustrasse 3, 14195 Berlin, Germany; 3Physikalisch-Technische Bundesanstalt Braunschweig und Berlin (PTB), Abbestrasse 2-12, 10587 Berlin, Germany

**Keywords:** GPCR states, Y2R, NMR, hyperpolarized xenon, MD simulation

## Abstract

G protein-coupled receptors can adopt many different conformational states, each of them exhibiting different restraints towards downstream signaling pathways. One promising strategy to identify and quantify this conformational landscape is to introduce a cysteine at a receptor site sensitive to different states and label this cysteine with a probe for detection. Here, the application of NMR of hyperpolarized ^129^Xe for the detection of the conformational states of human neuropeptide Y2 receptor is introduced. The xenon trapping cage molecule cryptophane-A attached to a cysteine in extracellular loop 2 of the receptor facilitates chemical exchange saturation transfer experiments without and in the presence of native ligand neuropeptide Y. High-quality spectra indicative of structural states of the receptor–cage conjugate were obtained. Specifically, five signals could be assigned to the conjugate in the apo form. After the addition of NPY, one additional signal and subtle modifications in the persisting signals could be detected. The correlation of the spectroscopic signals and structural states was achieved with molecular dynamics simulations, suggesting frequent contact between the xenon trapping cage and the receptor surface but a preferred interaction with the bound ligand.

## 1. Introduction

It is well accepted that transmembrane G protein-coupled receptors (GPCRs) exist in an equilibrium of multiple structural conformations during all steps of the signal transduction process. This conformational ensemble is modulated through interactions with ligands and transducers, which bind to the extracellular and intracellular receptor surfaces, respectively [1], and regulate many essential functions in the human body. The structure and thermodynamics of the individual receptor conformations can be summarized as a multidimensional surface consisting of energy wells and barriers (conformational landscapes) [2,3]. Knowledge of a receptor’s conformational landscape and how it is affected by ligand and transducer binding is of utter importance for the understanding of ligand action and GPCR-mediated signal transfer and provides the most rational basis for the design of therapeutics in case of malfunction.

In particular, NMR spectroscopy has proven its potential to decipher the structural dynamics of various receptor states in the signal transduction process [1,3,4]. The application of NMR spectroscopy in GPCR research is, however, challenging, as it requires milligram amounts of functional receptors embedded in a membrane environment. This issue could be resolved with *E. coli* expression of CXCR1 [5], BLT2 [6], GHSR [7], and Y2R [8] in inclusion bodies with subsequent in vitro folding. The NMR signals in such preparations, consisting of the receptor surrounded by detergents or phospholipids, are broadened and severely overlap because of the large molecular weight and comprehensive GPCR dynamics [9,10]. In this situation, even site-specific isotopic labeling using ^13^C and ^15^N is cumbersome. Particularly, the background of naturally abundant ^13^C atoms stemming from the lipid environment often renders the assignment of the receptor signals difficult. A strategy to overcome these problems is the tagging of receptors with a non-native label for background-free magnetic resonance investigations. Recent prominent applications use ^19^F probes at cysteine sites, as powerfully demonstrated in the analysis of the dynamic response of the ß_2_ adrenergic receptor upon agonist or antagonist binding [1,11]. A similar strategy is applied in electron paramagnetic resonance (EPR) spectroscopy, e.g., for the elucidation of the structural dynamics of bovine rhodopsin in various stages of its activation pathway [12]. While EPR can facilitate the efficient analysis of molecular motions due to the high inherent sensitivity, ^19^F-NMR requires much longer experimentation time and is restricted to one-dimensional spectra. As the various ^19^F signal positions and linewidths encode for the desired information, the approach is further susceptible to signal overlap. Crucial, as to the attachment of any NMR label, is the positioning of ^19^F to make it sensitive to the dynamic alterations in the receptor states upon activation or ligand binding events but not perturbing these well-balanced equilibria or the receptor folding in the membrane. Finally, the correlation of the ^19^F spectral features and the GPCR structural dynamics strongly improves with the availability of high-resolution structural data, possibly in combination with validating molecular dynamics (MD) calculations.

Given the number of constraints across various fields of expertise, the development of NMR methods for GPCR structural dynamics is challenging. Yet, a potentially promising route could be the use of isotope ^129^Xe as dynamics probe. Favorable properties are the good suitability for NMR due to its nuclear spin (1/2), the high sensitivity to the molecular environment due to the large range of accessible chemical shifts (>300 ppm in aqueous solution), and the easy and substantial solubility in physiological solutions (aqueous, as well as lipidic) for background-free measurements [13]. Moreover, by hyperpolarizing ^129^Xe gas (hyperpolarized ^129^Xe; hpXe) through spin-exchange optical pumping prior to its usage in the NMR experimentation in solution, its detection sensitivity can be boosted by orders of magnitude [14]. This would allow investigations to be conducted using reduced amounts of specimen. Of particular importance is the tendency of xenon to bind to hydrophobic micro-environments. Thus, small, synthetic host molecules can act together with hpXe guest atoms as a reporter system to sense biomolecular markers and processes, e.g., proteins in solution, receptors on cellular surface, or enzymatic activity [15,16,17]. Based on these assets, the feasibility to probe GPCR functional dynamics using hpXe NMR is investigated here. The approach is demonstrated using human Y2 receptor (Y2R), which is involved in a number of physiological processes, including food intake, neuroprotection, and circadian rhythm [18]. Its native ligand is the 36-amino acid neuropeptide Y (NPY) [19]. To facilitate monitoring using hpXe, a Y2R mutant with a free cysteine was covalently coupled to the synthetic, small xenon host molecule cryptophane-A (CrA). The spectroscopic signature of hpXe interacting with the Y2R-CrA conjugate alone as well as in complex with ligand peptide NPY was assessed to be informative in terms of receptor functional dynamics. The analysis of the NMR data was aided by MD simulations [20] of Xe bound to the Y2R-CrA conjugate alone or in complex with NPY in a membrane environment in aqueous solution.

## 2. Results and Discussion

### 2.1. Preparation of the Y2R-A202C-CrA Conjugate

As a template, we used a Y2R variant (Y2R-Cys-dpl), where the residues C58, C103, C151, C272 and C316 are changed to alanine or serine, respectively, depending on the expected hydrophobicity of the surroundings (Figure 1b). The only two remaining cysteines form the disulfide bridge between C123 in transmembrane helix 3 (TM3) and C203 in extracellular loop 2 (ECL2). This variant features pharmacological activity similar to that of the Y2R wild type [21]. We introduced an additional cysteine at position 202 (Y2R-A202C) directly neighboring the native C203 in ECL2. The Y2R-A202C mutant was used in previous EPR studies, and wild-type-like behavior was confirmed [22].

With yields of >10 mg/L medium, Y2R-A202C was expressed in *E. coli* in a fed-batch fermentation process [23] as inclusion bodies. The receptor protein was subsequently solubilized in sodium dodecyl sulfate (SDS), purified with immobilized metal affinity chromatography, and functionally reconstituted into a 1,2-dimyristol-sn-glycero-3-phophocholine (DMPC) bilayer [24], with yields of approximately 80% of the starting receptor material. The CrA-Cy3-mal construct (Figure 1a), bearing the hpXe binding molecule CrA, a linker of a Cy3 dye molecule and a maleimide group, was attached via maleimide reaction to Y2R-A202C (Figure 1b,c) at the free cysteine to form the conjugate Y2R-A202C-CrA.

Complete intramolecular disulfide bridge formation was verified by labeling free cysteines with *N*-[4(7-diethylamino-4-methyl-3-coumarinyl) phenyl]maleimide (CPM) and analyzed with fluorescence measurements (Figure 1d) [25]. By comparing the measured fluorescence intensities of the Y2R-Cys-dpl construct before and after folding [21], where all cysteines but the two for disulfide bridge formation were removed, and that of Y2R-A202C, containing one additional cysteine, intensity of approximately 2 a.u. of one free cysteine could be calculated. Considering the background fluorescence intensity of CPM of again 2 a.u., Y2R-A202C contained, after folding, one free cysteine for labeling with CrA. Further, in a fluorescence polarization assay with TAMRA-NPY (Figure 1e), the affinity of Y2R-A202C to its natural ligand, NPY, was confirmed to be in the same nanomolar range as it was measured for Y2R-Cys-dpl [24].

**Figure 1 molecules-28-01424-f001:**
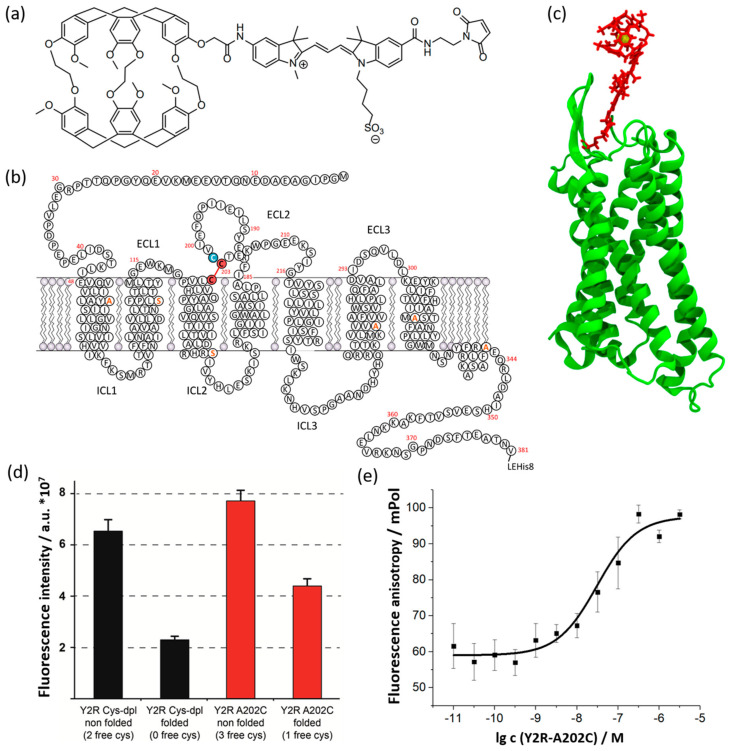
Representation of the Y2R-A202C-CrA construct. (**a**) Schematic chemical structure of CrA-Cy3-mal construct. (**b**) Sequence map of cytoplasmic Y2R-A202C. The cysteine introduced at position 202 (blue), the two cysteines forming the disulfide bond (red), and the replaced cysteines (orange) are marked. (**c**) Structure of Y2R [26] (green) with model of CrA-Cy3-mal (red) ) containing a xenon atom (yellow) bound to Cys202 (Y2R-A202C-CrA. (**d**) Results of the CPM assay for testing cysteine accessibility. High fluorescence intensities designate free cysteine residues. Y2R Cys-dpl (black), containing C123 and C203 for disulfide bridge formation, and Y2R-A202C (red), with the additional cysteine for CrA labeling, are compared. Before folding, all cysteines were free and accessible, while after folding, two cysteines formed a disulfide bridge. Three independent samples were measured. (**e**) Characterization of the ligand binding of Y2R-A202C using a fluorescence polarization assay with TAMRA-NPY [24]. Based on the saturation curve, a K_D_-value of 32 ± 12 nM was determined. Three independent samples were measured.

### 2.2. Xenon NMR

The hpXe sensing is based on host molecules that may interact, e.g., through suitable functionalization, with specific target structures, while at the same time, hpXe transiently bound to the host is detected to furnish evidence of the targeting event [15,16,17]. The approach is established using the host molecule CrA, a cage-like molecule that can enclose a single xenon atom for tens of milliseconds at ambient temperature [27,28]. Upon binding to CrA, a huge additional chemical shift (>100 ppm) is induced in the xenon atom with the precise extent depending on the molecular neighborhood of the host. The reversible CrA-hpXe complex formation is slow on the NMR timescale, enabling chemical exchange saturation transfer (CEST) techniques to further enhance detection sensitivity down to the micromolar concentration range and below [29,30]. To establish CEST-NMR on the CrA-hpXe host–guest system for the study of the Y2R-ligand interaction, CrA conjugation with the receptor was chosen. This approach allows Y2R to be studied alone or in complex with NPY, provided that CrA can be positioned to experience different local environments correspondingly. The Y2R-A202C variant offers, for conjugation, a free cysteine residue positioned next to the highly conserved disulfide bridge between TM3 and ECL2. This loop adopts a ß-hairpin structure and induces ligand interactions in the Y2R/NPY complex [31].

To test the suitability of this site for hpXe sensing, CrA was attached by means of thiol-maleimide “click” reaction to Y2R-A202C to form the conjugate Y2R-A202C-CrA. Between the cage molecule and the maleimide, linear cyanine-3 dye (Cy3) was incorporated as a spacer to allow some flexibility in the orientation and, at the same time, the distancing of CrA from the protein to be achieved (CrA-Cy3-mal construct, Section 2.1). In addition, the dye helped to exert some control over the preparation steps by means of color changes and could potentially be used for complementary fluorescence microscopy measurements. A sample of Y2R-A202C-CrA conjugate at 10 µM concentration embedded in DMPC bicelles and an equivalent sample incubated with NPY were investigated with CEST experiments (Section 4.3). Therein, the polarization of hpXe bound to the CrA moiety of the Y2R-A202C-CrA conjugate was depleted by means of selective RF irradiation, leading to a reduction in the signal of freely dissolved hpXe because of mutual exchange. Thus, monitoring the depletion of free hpXe during CEST amounts to the indirect detection of conjugate-bound hpXe, which, accumulated over time, is effectively amplified in comparison to direct detection [29,30]. Furthermore, by varying the saturating RF frequency, chemical shift information can be preserved, as evidenced by the resulting so-called z-spectra, which display signal intensity over RF irradiation frequency.

As a starting point, a buffer solution of Y2R embedded in DMPC bicelles was incubated with CrA-COOH at sufficient concentration to yield standard NMR spectra, where the signals of CrA-bound hpXe in both the aqueous and lipid phases were observed, beside the signal of freely dissolved hpXe (Figure 2a). The respective signal positions at 63.7 ppm, 77.3 ppm, and 196 ppm (not shown) are in full agreement with the reported data [32,33]. Furthermore, the linewidths of 320 Hz and 24 Hz for hpXe bound to CrA in the lipidic and aqueous phases, respectively, match the exchange rate of xenon for leaving the host molecule, as the natural width can be safely ignored against the exchange broadening, and the intensities reflect the preferential partitioning of hydrophobic CrA-COOH into the lipid phase [28,32,33,34,35]. In the next step, again, DMPC bicelles in buffer solution were investigated, this time incubated with the CrA-Cy3-mal construct. The z-spectrum (Figure 2b) comprises a single strong and broad resonance at the position expected for the lipid-embedded CrA moiety (standard spectra are due to the low sensitivity not being meaningful). As the resonance for the CrA moiety in the aqueous environment is absent in comparison with the free CrA case (Figure 2a), the construct may be even more hydrophobic than CrA alone and may have almost completely partitioned into the lipid phase. The spectral signature changes again and drastically when the CrA-Cy3-mal construct is not free but conjugated with Y2R embedded in bicelles. In Figure 2c,d, the z-spectra of hpXe bound to the CrA moiety for the Y2R-A202C-CrA conjugate alone and in complex with NPY are presented. In both cases, a peaked, broad CEST resonance line covers a spectral band exceeding the range of hpXe bound to free CrA in the lipidic and aqueous phases in the standard spectrum (Figure 2a) as well as that bound to the CrA moiety of the CrA-Cy3-mal construct in the z-spectrum (Figure 2b). Moreover, the line changes appearance in particular frequency spots whether hpXe interacted with the Y2R-A202C-CrA conjugate alone or in complex with NPY (Figure 3). The structured line shape for both samples and the frequency-specific deviations suggest the broad resonance in either spectrum to be formed by the superposition of a small number of individual resonances. Indeed, the best fit of the experimental data could be obtained by means of the superposition of five signals for the bare conjugate and six signals when in complex with NPY (Figure 2c,d).

The model function for CEST resonance due to hpXe exchange with a CrA host molecule is described as an excellent approximation using an exponential Lorentzian, with the product of exponentials for a superposition [36,37].
(1a)$$S\left(x\right)=A\;\underset{i}{\prod }e^{-B_{i}\frac{a_{i}^{2}}{a_{i}^{2}+{\left(b_{i}-x\right)}^{2}}}$$
(1b)$$≅A\;e^{-B_{3}\frac{a_{3}^{2}}{a_{3}^{2}+{\left(b_{3}-x\right)}^{2}}}\;\left(1-\underset{i\neq 3}{\sum }B_{i}\frac{a_{i}^{2}}{a_{i}^{2}+{\left(b_{i}-x\right)}^{2}}\right)$$
where index *i* counts the number of resonances fitted to the data, running here to five or six for the conjugate alone or in complex with NPY, respectively. The leading amplitude, *A*, defines the baseline in the z-spectrum. Parameter *b_i_* is the Larmor frequency (position) of the *i*-th resonance, and *x* denotes the applied RF irradiation frequency, which was systematically varied in the CEST experiment. It is common to express these frequencies not in absolute terms but, as for any NMR spectrum, in ppm, referenced here to the xenon gas resonance (0 ppm). Parameters *B_i_* and *a_i_* are the amplitude and half width of the underlying Lorentzian, respectively. The parameters of the individual resonances, conveniently numbered in the order of appearance upfield to the signal of freely dissolved hpXe (at 196 ppm), are listed in Table 1. The argument of the exponential of each fitted resonance is of the largest magnitude for *x* = *b_i_*, namely, *B_i_*. According to Table 1, for resonances 1, 2, 4, 5, and 6, *B_i_* < 0.2 holds, while for resonance 3, *B*_3_ > 1. These numbers are also upper bounds for the complete Lorentzian argument when *x* deviates from *b_i_*. For resonances 1, 2, 4, 5, and 6, with these bounds being much less than unity, the respective exponentials in the model can be approximated by the terms linear in the Lorentzian (Equation (1b)), meaning that the signals in the z-spectra—with the exception of resonance 3—are Lorentzians reflected at the baseline of amplitude *A*. However, despite this simplification, straightforward spectral analysis is impeded, as parameters *B_i_* and *a_i_* are intricate quantities reflecting the mechanism of CEST.
(2a)$$B_{i}=\;t\;k_{on,i}\frac{\omega_{1}^{2}}{\omega_{1}^{2}+k_{off,i}^{2}+k_{off,i}R_{2,i}}$$
(2b)$$a_{i}=\;\sqrt{\omega_{1}^{2}+k_{off,i}^{2}+R_{2,i}^{2}+\frac{R_{2,i}}{k_{off,i}}\left(\omega_{1}^{2}+2k_{off,i}^{2}\right)}$$
where *t* is the period of irradiation; *ω*_1_ denotes the xenon nutation frequency in the applied RF; for either resonance, *R*_2,*i*_ is the transverse relaxation rate of conjugate-bound hpXe; and *k_on_*_,*i*_ and *k_off_*_,*i*_ are the rate coefficients for hpXe entering and leaving the complex with the host, respectively [36].

At first glance, the line shape of the superposition is more accentuated for the complex with NPY than that for the Y2R-A202C-CrA construct alone. This finding may be related to a rigidification of the molecular structure when the ligand was bound, which is also reflected by the systematically reduced uncertainties in the fitted signal parameters in the latter case (Table 1).

In addition, the linewidths of hpXe in the Y2R-A202C-CrA conjugate alone or in complex with NPY for all resonances listed in Table 1 indicate a strong impact of relaxation, as the respective values are generally above the exchange widths of 24 Hz and 320 Hz for free CrA-COOH in aqueous and lipid environments, respectively (Figure 2a). More specifically, the RF amplitude applied in the CEST experiments of $\omega_{1}=2\pi \;99\;\mathrm{Hz}$ is much stronger than the exchange rate for CrA-COOH in the aqueous environment and about twice the one for CrA-COOH in lipids. The respective width parameter (Equation (2b)) thus becomes $a_{i}=\;\sqrt{\omega_{1}^{2}+R_{2,i}^{2}+\frac{R_{2,i}}{k_{off,i}}\omega_{1}^{2}}$, displaying the possible dominance of *R*_2_ processes. This is not surprising, because the CrA-Cy3-mal construct was rather rigid due to the planar structure of the dye. Consequently, the high molecular weight of the Y2R-A202C-CrA conjugate embedded in bicelles drastically affected the relaxation of bound hpXe through the corresponding slow rotational tumbling. Particularly enhanced were the dipolar relaxation induced by the hydrogens lining the binding cavity due to their proximity to the enclosed xenon and the relaxation induced by chemical shift anisotropy due to the large polarizability of the xenon electron shell [38]. Furthermore, contributions to the resonance width by fluctuations between lowly populated conformations of the conjugate are possible, particularly in case of ligand-free Y2R-A202C-CrA.

Concerning the individual signals in the superposition, the strong central resonance 3 and the downfield shifted resonances 1 and 2 are virtually in identical positions in the absence and presence of NPY. These signals are thus likely not indicative of a state change upon NPY binding. Resonance 3 dominates the superposition at position 77.5 ppm, which matches well the signals of hpXe exchanging with free CrA-COOH (Figure 2a) or with free CrA-Cy3-mal (Figure 2b), both in the lipid environment. Thus, resonance 3 can be assigned to hpXe interacting with the CrA moiety in the conjugate in the lipid environment. During sample preparation, the CrA-Cy3-mal construct was added in double excess to the receptor mutant embedded in bicelles. Due to the hydrophobicity imparted by the cage and the dye, CrA-Cy3-mal may have dominantly partitioned into the bicelles, with especially its CrA headpiece surrounded by lipids, and material may have remained there after washing. Noteworthy, due to the enhanced xenon exchange rate of the CrA moiety in lipid compared with aqueous solutions (320 Hz vs. 24 Hz), the CEST effect, and thus resonance 3, was appropriately amplified. Therefore, resonance 3, albeit surmounting the other signals, may be considered unrelated to Y2R-A202C and its binding activation. Resonances 1 and 2, although located in very similar positions with and without NPY-bound Y2R-A202C-CrA, are shifted downfield by ~27 ppm and ~14 ppm (b1 and b2), respectively, from the position associated with CrA in lipids (resonance 3) and even more in the aqueous environment. Large chemical shifts of that order in comparison with xenon bound to free CrA-COOH can be induced either by direct proximity of a de-shielding molecular moiety or its proximity to CrA acting as a transducer [39]. Resonances 1 and 2 are thus likely representative of structural states of a well-confined CrA moiety, irrespective of the presence of the ligand. Here, further analysis using complementary experimental or computational structural data is required to sort out the conformational details.

In the z-spectra of hpXe in apo and NPY-bound Y2R-A202C-CrA (Figure 2c,d), marked differences are displayed by resonances 4, 5, and 6. Chemical shift *b*_5_ for Y2R-A202C-CrA in complex with NPY matches well the one in the control spectrum of hpXe bound to free CrA-COOH in aqueous solution (Figure 2a). For Y2R-A202C-CrA alone, *b*_5_ is shifted further upfield by an amount of about the summed standard errors of both positions. In addition, the large linewidth, twice that of the already broad signal of the complex, renders the position comparatively indefinite. Therefore, resonance 5 may be assigned to conjugated but fairly unrestricted CrA in the water environment for the complex but possibly represents structural interference of the CrA moiety when the conjugate is unligated. Resonance 4 is shifted downfield by ~3 ppm or ~7 ppm compared with free CrA-COOH in aqueous solution for Y2R-A202C-CrA alone and in complex with NPY, respectively. These shifts are similar in size to what has been reported for CrA interacting with protein surfaces [16,17,38] and were presumably affected by a stronger polarization when hpXe approached the hydrogen and carbon atoms lining the cage interior, i.e., when the volume accessible to the xenon atom shrank [39]. The uncertainties in the respective signal positions and the large linewidth in comparison with their separation rendered the effect of NPY binding on resonance 4 rather minor. In addition, assuming that hpXe exchange took place at equal rates in aqueous solution, the state represented by resonance 4 is significantly less populated than the state represented by resonance 5 in the respective spectra due to smaller signal intensities (signal area). In contrast, resonance 6 may represent a state uniquely associated to NPY in complex with Y2R-A202C-CrA, as no signal is present around that chemical shift for the conjugate alone. For resonance 6, present only for the complex, an unusually strong upfield shift of ~14 ppm compared with free CrA-COOH in the aqueous environment occurs. A small contribution to such large shifts may have stemmed from an enlarged volume of the binding cavity compared with free CrA-COOH due to the replacement of the carboxyl group at the rim of CrA-COOH by the Cy3 linker [39]. Much stronger shifts, however, can be induced by charges affecting the polarization of bound xenon. For example, head groups of phospholipids forming the bicelles or charged side chains located on the receptor may be in direct proximity of a bound xenon atom or at least to the CrA moiety that than transduces polarization changes to the hpXe atom in the interior. Again, only in combination with further structural data, the details of such interaction and their relation to hpXe-NMR may be revealed.

The presented CEST data of the Y2R-A202C-CrA conjugate alone and in complex with the NPY ligand were qualitatively analyzed in terms of signal position and width. A semi- or fully quantitative evaluation based on fitting the model using the explicit expressions in Equation (2a,b) would require strong resonances 1, 2, 4, 5, and 6 for low numerical uncertainties. While, principally, the signals could be enhanced with a stronger CEST, e.g., through more powerful or longer RF irradiation, the large overlap with resonance 3 would persist and impede this approach. One must also keep in mind that the spectral signature of the overall superposition of resonances 1 to 6 changes among preparations when the equilibrium of receptor-bound versus free constructs varies and, similarly, when the equilibrium of bound to free ligands shifts. Consequently, a quantitative analysis of the spectral parameters is prone to a basic bias reflecting variations in sample preparation.

### 2.3. MD Simulation

To interpret the experimental observations of multiple peaks with varying positions and intensities, we performed MD simulations of the apo Y2R-A202C-CrA and Y2R-A202C-CrA/NPY systems. Since contacts between the cage and the protein or membrane are relatively long-lived, we ran 30 simulations per system at different initial velocities to sample all contact sites. A representative starting conformation is shown in Figure 4.

For trajectory analysis, the instances of contact between the xenon atom inside the cage and sites outside of the cage were quantified. A contact instance was counted as such if xenon was within an 8.5 Å distance from another amino acid or molecule in a given frame (Appendix A). In Figure 5a,b, all amino acid contact instances of xenon are visualized for the two systems. For apo Y2R-A202C-CrA, contact with all extracellular loops and the N-terminus of Y2R was frequently observed, especially at the N-terminus and in ECL2. For the Y2R-A202C-CrA/NPY complex, much fewer xenon contact sites were observed, mostly related to NPY contact instances. Moreover, the sites in ECL2 and at the N-terminus that were most frequently subject to contact in the apo state were also observed in the NPY-bound state, whereas the contact instances that were less frequently observed in the apo state were not seen at all in the NPY-bound state. The reason for this could be the steric restriction of the movement of CrA by NPY, which occupied the ligand binding pocket in the central position.

Since the frequency of contact, normalized separately for each system, cannot be used for comparison between the apo and NPY-bound states, contact probabilities were calculated instead. In Figure 5c, contact cases are discriminated between “just lipid” (in one frame, xenon only comes into contact with lipid molecules), “just protein” (in one frame, xenon only comes into contact with either Y2R or NPY), “lipid and protein” (in one frame, xenon comes into contact with lipid molecules and either Y2R or NPY simultaneously), and “no contacts” (in one frame, xenon does not come into contact with any other amino acid or lipid molecule). According to this diagram, it becomes apparent that in the presence of NPY, xenon came into contact with proteins almost exclusively, while in the apo state, it also frequently came into contact with lipids or often experienced no contact at all (indicating that the cage was in aqueous solution). In this figure, all protein contact instances fall either in the category “just protein” or “lipid and protein”. However, multiple protein contact sites can be observed in Figure 5a,b. To also quantify these contact instances and facilitate the comparison between the two systems, the contact probabilities of these sites were calculated. One difficulty in calculating these contact probabilities is to treat all contact sites similarly, although the cage might experience different flexibility at different sites and, depending on the shape of the contact site surface, it might come into contact with a different number of amino acids simultaneously. Therefore, we decided to quantify the contact probability of each site by averaging the contact probabilities of the three amino acids most often subjected to contact at each site. We decided to consider the following contact sites, each as a single cluster: NTER, ECL1 (at the end of helix 2), ECL2_1 (closer to helix 4), ECL2_2 (closer to helix 5), ECL3, and NPY NTER (the flexible part of NPY) as well as NPY helix (the part forming a stable helix) in the Y2R-A202C-CrA/NPY complex, as listed in Table 2.

The resulting contact probabilities for each site are shown in Figure 5d. While NPY is emerging as the main contact site for xenon in the respective setup, some contact sites in apo Y2R-A202C-CrA are now disappearing completely. However, the most frequent contact sites in the apo state were also observed with comparable frequency in the presence of NPY. This leads us to the conclusion that some more rarely visited sites in the apo state were blocked by the presence of NPY. In contrast, other sites that were well accessible in the absence of NPY were still similarly accessible. In addition, while the cage was often exposed to solvent in the apo state, this was reduced significantly in the presence of the bulky NPY. This can be most likely attributed to two effects: First, the cage and the xenon inside are relatively apolar and thus might favor contact with amino acids over polar water, in particular since the linker and cage can remain in a more stretched conformation due to the location of NPY, which might be favorable due to the limited flexibility of the linker. Second, NPY occupies so much space above Y2R that there are simply much fewer conformations in which the cage does not touch any amino acid.

To investigate the influence of CrA-Cy3-mal on receptor dynamics, we conducted an additional 30 reference MD simulations per system, each of 1 microsecond in length, for Y2R in the absence and presence of NPY. In these simulations, the CrA-Cy3-mal construct containing the xenon atom was removed from the Y2R-A202C-CrA conjugate, so that only the Y2R-A202C mutant remained, thus allowing the equivalent simulations of the complete conjugate, i.e., Y2R-A202C-CrA, to be compared with the construct present. For all simulations of Y2R-A202C and Y2R-A202C-CrA with or without NPY, the order parameters were calculated for each amino acid Cα-H bond (Figure 6). Each order parameter is indicative of the amplitude of motion of the corresponding amino acid within the receptor or receptor/NPY complex, respectively, where the motion of the receptor or receptor/NPY complex, respectively, as a whole has been removed. The averages of all order parameters of the receptor apo state were rather similar, irrespective of CrA-Cy3-mal conjugation (without construct, 0.852; with construct, 0.842), and similar for the NPY-bound state (without construct, 0.869; with construct, 0.824). In the latter case, also for ligand NPY, the average values were very similar (without construct, 0.760; with construct, 0.731). When analyzing the order parameters of each amino acid individually, one has to be careful not to overinterpret small differences, since the conformational sampling of GPCRs is very challenging even for long trajectories such as the present cumulative 30 microseconds (e.g., any difference in intracellular loops is most likely random). Nevertheless, some patterns appeared. For all four trajectories, the order parameters of any specific acid in the helices of Y2R, in the intracellular loops as well as the C-terminus, were very close, which also held for the bound ligand, NPY. On the extracellular side, however, the presence of the construct seemed to increase the amplitude of motion (in particular for ECL3). However, the reason for this apparent increase in amplitude could also be that the construct reduced the correlation time of the motions but not their amplitude, which would have led to the improved sampling of conformations in the presence of the construct and thus apparent lower order. From the MD simulations alone, we could not discriminate between these two effects. Overall, however, the structural dynamics of Y2R seemed to be mostly unaffected by the presence of the cage, with the possible exception of ECL3, which seemed to become more flexible in the presence of the cage.

### 2.4. Signal Assignment

Comparing the data from the NMR measurements and the following MD simulations, conclusions on signal assignment as a working hypothesis can be drawn. In the NMR spectra, six signals for NPY-bound Y2R can be detected, one more than for the apo state. Most likely, this signal, resonance 6 at 49.5 ppm, can be attributed to a CrA/NPY contact, which was also detected in the MD simulation with a high probability. In contrast to the NMR spectra, the MD simulation detected fewer contact instances in the NPY-bound state than in the apo state. An explanation is that in the “real” sample of the Y2R-A202C-CrA/NPY complex, not all receptors were occupied by NPY; therefore, signals for both the apo and NPY-bound states were detected.

The dominating signal in all NMR spectra, resonance 3, appears at 77.5 ppm and, according to the hpXe control spectrum (Figure 2a,b), can be assigned to phospholipid contact. However, MD simulations detected only a low probability of lipid contact. Apparently, in spite of extensive washing, there was a large amount of non-bound CrA-Cy3-mal left in the bicelles (Figure 2b); in addition, cage moiety of Y2R-A202C-CrA likely came into contact with bicelles stacked on top of the receptor. This stacking effect was previously shown in negative-stain EM images of this sample preparation [24] but cannot be easily replicated with MD simulations. Instead, the detected CrA in solution with no contact could have well experienced lipid contact with opposing bicelles in the NMR sample.

Two CrA/Y2R contact instances showed only minor differences between the apo and the NPY-bound states in the MD simulations: the contact with the Y2R N-terminus and that with the site of ECL 2, which is close to TM4. This is in agreement with the cryo-EM structure of the NPY/Y2R complex [26], where both sites show no direct contact with NPY. The two far downfield signals in the NMR spectra at 92 (resonance 2) and 104 ppm (resonance 1) also show only very minor spectral differences due to ligand binding, and their assignment to the two contact sites might be obvious, although speculative. Nevertheless, neither of the signals and neither of contact instances was sensitive to NPY binding. While waiting for further experimental clarification, such strong downfield shifts may partly be explained by ring current effects, as CrA came into contact with W207 in ECL2, or by contact with charged amino acids, for example, K45 at the N-terminus.

In contrast, NMR resonances 4 and 5 were sensitive to NPY binding, in addition to signal 6, appearing at 49.5 ppm. Resonance 5 changed its position at 58 ppm in the apo state to 63 ppm in the NPY-bound state. In addition, its amplitude and its linewidth changed upon the addition of NPY. The MD simulations suggest contact between CrA and all ECLs of Y2R in the apo state, some of which were diminished in the NPY-bound state. The ligand sterically hindered contact with them, and instead, CrA came into contact with NPY at its hydrophobic helix. This resonance 5 could well be a superposition of these different ECL contact instances, which changed their quantitative distribution upon NPY binding and thus changed the position and linewidth of the signal. This assumption is supported by the fact that the ECLs and thus the CrA contact instances were exposed to the buffer solution, which is in agreement with the determined chemical shift for CrA in solution (Figure 2a), and that resonance 5 is of comparably large width for apo and NPY-bound Y2R. Furthermore, more intense contact between the CrA moiety and the ECLs could have affected the cage conformation and thus induced the downfield shifts manifested in resonance 4 as discussed above (Section 2.2). The difference in resonance 4 for apo and NPY-bound receptors of ~3 ppm may indicate a secondary effect of NPY binding due to stronger deformation in the CrA cage through ECL contact.

In the NPY/Y2R complex, most contact instances of CrA were seen with the NPY helix and NPY N-terminus. Both sites were close in space, and quite often, the cage was in contact with both simultaneously. Explanations are that NPY sterically hindered CrA from coming into contact with ECL1, as observed in the apo state, and the hydrophobic helix of NPY caught CrA otherwise pointing to the solution in the apo state. Surprisingly, we detected significantly lower order parameters for ECL3, indicative of higher mobility of Y2R-A202C-CrA in the apo as well as in the NPY-bound states (Figure 6). Possibly, to some extent, the CrA-Cy3-mal construct replaced the ECL3 contact with NPY or ECL2 in the apo state, which was also seen in the experimental structures [40].

## 3. Conclusions

The understanding of the conformational dynamics of GPCRs and changes in their population distribution upon interaction with extra- or intracellular ligands and transducers enables compounds modulating the various signal transduction pathways to be identified. Here, we introduce the usage of hpXe NMR for detecting Y2R conformational states. To our knowledge, in GPCR research, xenon has only been used to detect hydrophobic pockets in ß1-AR so far [41].

We attached the xenon trapping cage molecule CrA to a free cysteine at the edge of the extracellular binding pocket of Y2R and proved wild-type-like activity of the variant. By applying NMR measurements of hyperpolarized ^129^Xe, five to six isolated signals were detected in the apo state or in the NPY-bound state of Y2R, respectively, where three of these signals were recognized to be sensitive to ligand binding. Using MD simulations, we identified and quantified the contact between CrA and all ECLs of Y2R in the apo state, which were partially impeded by ligand binding. We also identified contact instances at the N-terminus and ECL2 of Y2R, which are independent of NPY. Further, we quantified contact probabilities in different environments of CrA and suggested the assignment of NMR signals.

The high sensitivity of hpXe in our sample preparation and the outstanding dispersion of signals of over 50 ppm demonstrate the promising potential of the techniques introduced here for deciphering the conformational landscape of Y2R. Particularly, hpXe NMR approaches for further quantitative evaluations, down to the parameters governing xenon exchange with the conjugate, are available [37,42]. Specifically, the determination of the concentration of the conjugate in its different states, i.e., the fractional populations, could be very helpful in the exploration of receptor activity. Along this route, a number of varied experimental schemes may be applied, e.g., using construct architectures of different linker length or linker flexibility in Y2R activation by different ligands, individually or in combination, at the extra- and intracellular receptor binding sites [22] to ultimately arrive at a quantitative and comprehensive description of Y2R activity. Moreover, complementary solution- or solid-state NMR spectroscopy using specific isotopic labeling may be undertaken to verify contact sites and MD employed for refining the insights into Y2R structural dynamics.

## 4. Materials and Methods

### 4.1. Sample Preparation

Y2R-A202C, C-terminally flanked with a poly-8-His-tag, was expressed in *E. coli* Rosetta^TM^ (DH3) pLysS strains as inclusion bodies during a fed-batch fermentation run in defined minimal salt medium. The purification of inclusion bodies, protein solubilization in SDS and DTT containing buffer, and the His-tag-based IMAC purification of the unfolded receptor proteins were performed according to the well-established standard protocol [23]. The functional reconstitution of the receptor into non-isotropic DMPC bicelles was performed as previously described [24]. CPM and fluorescence polarization assays were performed as previously conducted [22]. After functional reconstitution, Y2R-A202C was incubated with 10× molar excess of CrA-Cy3-mal overnight at 4 °C and pH 7 to form Y2R-A202C-CrA. Non-bound CrA was removed with eight cycles of pelleting the receptor in bicelles by centrifugation (8 min, 4 °C, 21,500× *g*) and re-solubilizing the pellet in 50 mM sodium phosphate (pH 7). The receptor concentration was set to 10 µM for NMR measurements.

To prepare the Y2R-A202C-CrA/NPY complex, Y2R-A202C-CrA was incubated with 2× molar excess of pNPY, produced with solid-phase synthesis [19] overnight at 4 °C. Non-bound NPY was removed by pelleting the receptor and resolubilizing in fresh buffer (50 mM sodium phosphate; pH 7), again to a concentration of 10 µM.

### 4.2. CrA Functionalization

Cryptophane-Cy3-COOH. CryptophaneA (50.0 mg, 53 µmol) was dissolved in dry dichloromethane (1 mL), and thionyl chloride was slowly added. The reaction mixture was refluxed for 1 d and then concentrated under reduced pressure. CryptophaneA acid chloride was used without further purification. 

To a solution of CryptophaneA acid chloride (22.0 mg, 23 µmol) and NH2-Cy3-COOH (5.0 mg, 9.3 µmol) in (225 µL) DMF, DIPEA was added (5 µL, 3.5 mg, 28 µmol). The reaction mixture was stirred at 50 °C for 1 h and at 20 °C for 20 h. The solvent was removed under reduced pressure, and the crude product was purified using column chromatography (SiO_2_, dichloromethane 100% to dichloromethane/methanol at 80:20) to obtain CryptophaneA-Cy3-COOH (19.2 mg, 71%) as a red solid. ^1^H NMR (400 MHz, Methanol-d_4_) δ 8.73–8.58 (m, 1H), 8.35–8.10 (m, 4H), 7.96 (d, 1H), 7.78 (s, 1H), 7.45 (dd, 2H), 7.17–6.86 (m, 10H), 6.73–6.58 (m, 2H), 4.90–3.47 (m, 39H), 3.26–3.01 (m, 9H), 2.19 (s, 4H), 2.05–1.89 (m, 12H); MS(ESI^+^): *m/z* = 1456.5672C_84_H_88_N_3_O_18_S^−^ (calculated = 1456.5633).

CryptophaneA-Cy3-mal. A solution of CryptophaneA-Cy3-COOH (19.0 mg, 13 µmol), *N,N′*-dicyclohexylcarbodiimide (8.0 mg, 39 µmol), and N-hydroxysuccinimide (4.5 mg, 39 µmol) in dichloromethane (700 µL) was stirred at 20 °C for 24 h. The reaction mixture was filtered, and NHS ester was used without further purification. MS(ESI^+^): *m/z* = 1577.5701 C_88_H_90_N_4_NaO_20_S^+^ (calculated = 1577.5761).

CryptophaneA-Cy3-NHS (20.3 mg, 13 µmol), *N*-(3-aminopropyl)maleimide hydrochloride salt (14.0 mg, 78 mmol), and *N,N*-diisopropylethylamine (13.2 µL, 10 mg, 78 µmol) were dissolved in dichloromethane (700 µL) and stirred at 20 °C for 5 d. The reaction mixture was washed with water (10 mL), and the aqueous layer was extracted with dichloromethane (2 × 10 mL). The combined organic layers were dried over MgSO_4_, filtered, and concentrated under reduced pressure. The crude product was purified using automated column chromatography (4 g of SiO_2_, dichloromethane 100% to dichloromethane/methanol at 85:15), and CryptophaneA-Cy3-mal (12.4 mg, 60%) was obtained as a red solid. ^1^H NMR (400 MHz, Chloroform-d) δ 9.61–9.18 (m, 1H), 8.65–7.42 (m, 6H), 7.20–7.11 (m, 2H), 6.89–6.58 (m, 14H), 4.85–2.90 (m, 52H), 2.48–1.40 (m, 18H); MS(ESI^+^): *m/z* = 1603.6100 C_90_H_93_N_5_NaO_19_S^+^ (calculated = 1603.6112).

### 4.3. NMR Experiments

The hpXe NMR experiments were conducted using a wide-bore 7 T spectrometer (Bruker Corporation, Billerica, MA, USA) equipped with a 10 mm double-resonant probe for the excitation and detection of X-nuclei (tuned to ^129^Xe) and ^1^H. Samples were measured at a temperature of 298 K to enable the comparison to reference data used for spectral assignment and analysis to be performed [35,36,37]. The NMR sample tube in the center of the NMR magnet was gastight-connected to a tube set up for the delivery of hyperpolarized xenon in a gas mixture from a distant xenon polarizer. The home-built polarizer operated in continuous mode at partial pressures of 0.02 bar, 2.78 bar, and 0.2 bar for xenon (^129^Xe in natural abundance of 26%), helium, and nitrogen, respectively [31]. Xenon polarization was ~18%. By means of a two-way valve, the gas stream could alternately bypass or be led through the sample. All timing for the control of the gas flow was implemented through TTL signals triggered in the NMR pulse sequence. For xenon dissolution in the sample liquid, the gas mixture was vigorously bubbled for 10 s, and the sample came to rest within a subsequent delay of 10 s. After a few cycles, the solution was completely saturated with xenon, with each additional cycle generating a stable replenishment of the hpXe concentration in the liquid. Anti-foaming agent was added to the samples for the prevention of excessive foaming. For the standard spectrum (Figure 2a), 32 scans were averaged, and for each one, the bubbling cycle was followed by a non-selective π/2 excitation pulse of 25 μs in duration and subsequent signal detection. For the acquisition of CEST data, after the bubbling cycle, a weak cw-irradiation (xenon nutation frequency: 2π 99 Hz) was applied for 20 s at given constant irradiation frequency to RF-saturate encaged xenon and was followed by RF excitation and detection of the signal of freely dissolved hpXe. To generate the z-spectrum, the irradiation frequency was stepped through scan by scan in a range from 5 kHz downfield to 13 kHz upfield with respect to the free xenon Larmor frequency, with a step size of 200 Hz. The FID in each scan was Fourier-transformed after apodization using an exponential filter function of 5 Hz. The single-signal spectra were phase- and baseline-corrected to absorption mode. The signal in each spectrum was integrated over an area of 7 ppm centered on the maximum signal amplitude. All data processing was performed using Topspin and Mnova software (Bruker). The CEST data obtained in this manner (Figure 2 and Figure 3) were further evaluated by fitting, with exponential Lorentzian functions, the theoretical line shape of the resonances [35,36,37] using Qti-plot software. The uncertainty in the values obtained for the different parameters is the standard error of the fitting. The measured z-spectra for Y2R-A202C-CrA alone or in complex with NPY, in Figure 2 and Figure 3, could only be adequately described when 5 or 6 resonances, respectively, were assumed in the fitted model.

### 4.4. MD Simulations

Two different systems were simulated: apo Y2R-A202C-CrA and Y2R-A202C-CrA/ NPY. The structure of Y2R was taken from RCSB PDB (7X9B) [31]. All non-receptor parts were removed, and the sequence was mutated to match the experiment. Receptor cavities were filled with water using dowser [43].

The CHARMM36 force field [44] was employed for lipids and proteins. Force field parameters and partial charges of the CrA molecule were created using CGenFF [45]. We tried to replace all bonded parameters that were above a penalty of 2 with existing parameters from the CY3R residue of the CHARMM force field, such that only very few parameters with any significant penalties remained. The non-bonded parameters for the xenon atom were taken from the literature. Three different publications with sometimes rather different parameters were found [46,47,48]. Since those publications also contained parameters for neon and, in one case, also helium, those were compared to the existing values in the CHARMM force field, and the publication with the best agreement was chosen [46].

System setup was conducted using CHARMM-GUI [49,50,51,52,53,54]. The N- and C-termini of Y2R were capped with the ACE and CT1 patches and those of NPY were capped with NTER and CT2 from the CHARMM force field, respectively. All residues were kept in the standard protonation states, with the exception of the highly conserved Asp96 and Asp147, which were protonated. Both systems contained 100 DMPC lipids in each leaflet. Since the use of 50 mM Na_3_PO_4_ as in the experiments led to its precipitation, 150 mM NaCl was used to achieve the same ionic strength instead. The box size was in the order of 85 × 85 × 135 Ǻ. This was chosen such that no part of the protein or the cage could reach through the periodic boundary to touch a periodic copy of the protein (see Figure 4, which represents exactly one periodic cell).

The simulations were run in the NPT ensemble at a temperature of 298 K and a pressure of 1.013 bar using GROMACS 2022. Particle-mesh Ewald was used to treat electrostatic interactions, using a cut-off distance of 10 Å. Bonds involving hydrogen were constrained with LINCS [55] to achieve a time step of 2 fs. Each system containing about 97,000 atoms was energy-minimized with the steepest descents algorithm and 1000 kJ mol^−1^ nm^−1^ as the threshold. All systems were equilibrated with harmonic positional restraints applied to lipids and Cα atoms of the protein and ligand that were sequentially released in a series of equilibration steps. For each system, 30 copies were simulated for 1000 ns each. The first 100 ns were considered equilibration and thus not used for analysis. Representative RMSD plots are given in Appendix A. For data evaluation, the probability of contact between the xenon atom and each amino acid (only heavy atoms were used to determine contact) was evaluated, where distances below 8.5 Å were counted as a contact instance. Data for other cut-off distances are shown in Appendix A. In addition, reference MD simulations in the absence of CrA (but in the presence of the A202C mutation) were conducted. Everything else was identical, and again, Y2R in the absence and presence of NPY was simulated for 30 × 1000 ns. Internal order parameters were calculated for all systems by aligning Y2R to its starting structure to remove overall receptor reorientation [20]. The calculation of the order parameters was not conducted using the 2nd Legendre polynomial, since this would have required receptor rotation around the membrane normal to be averaged out, which is not the case for such large molecules on the NMR time scale. Instead, order parameter calculation followed published procedures [56].

## Figures and Tables

**Figure 2 molecules-28-01424-f002:**
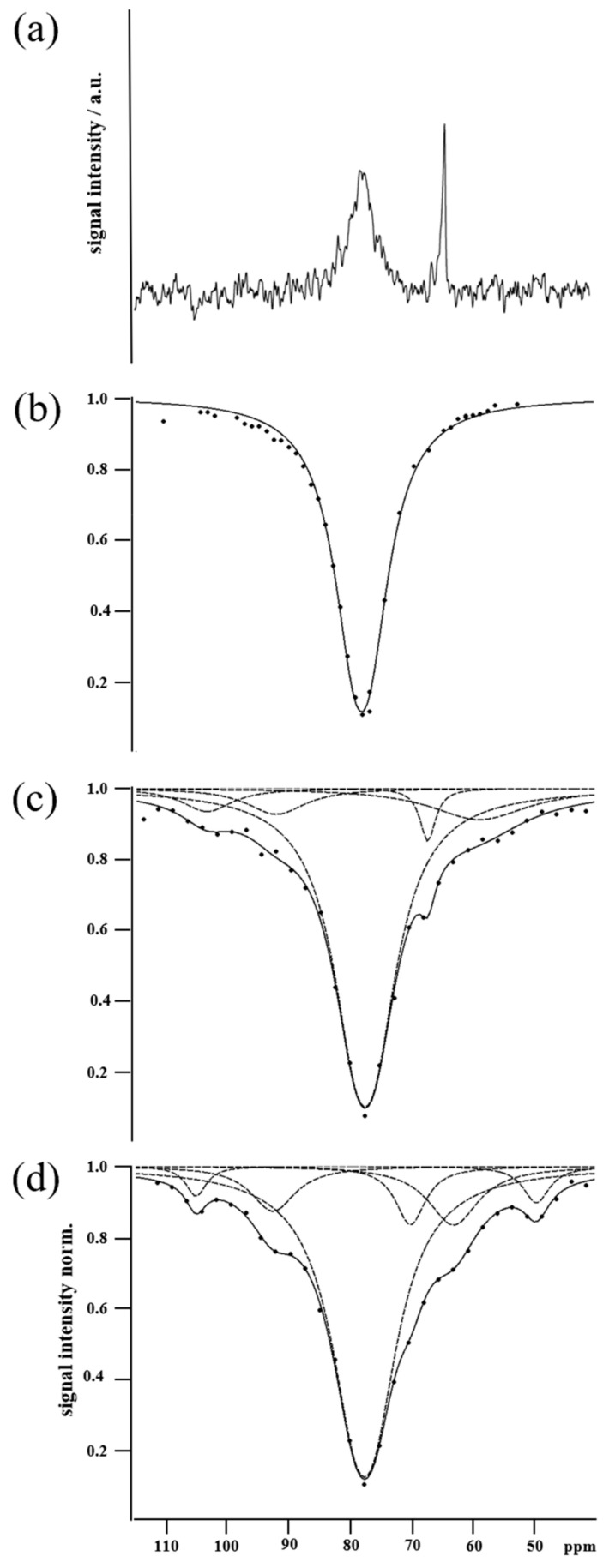
(**a**) Standard hpXe spectrum of bicelle-embedded Y2R in buffer saturated with CrA-COOH (excerpt, signal of freely dissolved hpXe at 196 ppm not shown). (**b**) Z-spectrum of hpXe bound to the CrA-Cy3-mal construct in a buffer solution of DMPC bicelles. The continuous line is the fit of a single Lorentzian model function to the experimental data (dots). (**c**) Z-spectrum of hpXe bound to Y2R-A202C-CrA conjugate. (**d**) Z-spectrum of hpXe bound to Y2R-A202C-CrA conjugate in complex with NPY. In (**c**,**d**), the continuous lines (black) are the fits of the model function to the data (black dots) as superpositions of individual resonances (gray dotted lines).

**Figure 3 molecules-28-01424-f003:**
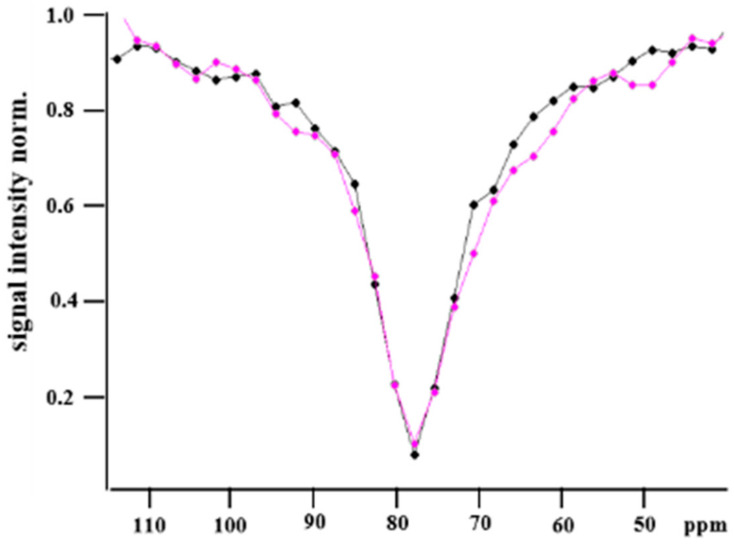
CEST resonance of hpXe bound to Y2R-A202C-CrA conjugate alone (black) and in complex with NPY (magenta). Continuous lines connect experimental data (dots) as a guide for the eye.

**Figure 4 molecules-28-01424-f004:**
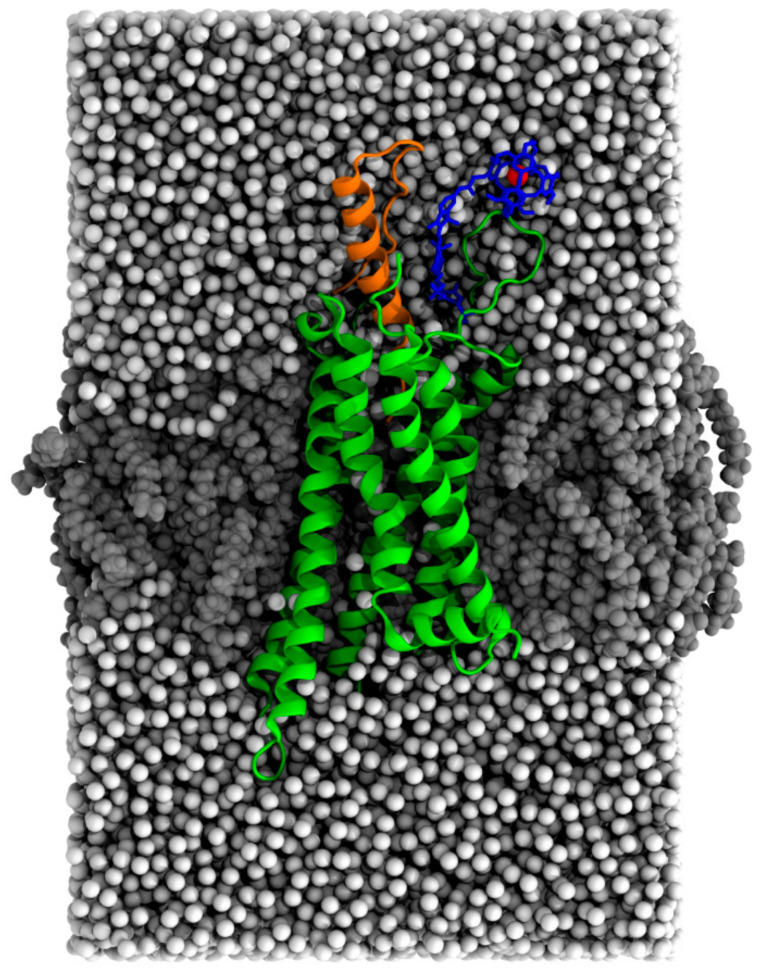
Snapshot (97,116 atoms) of the equilibrated Y2R-A202C-CrA system in complex with NPY after all restraints were released for 5 nanoseconds. Y2R is shown in green, NPY in orange, CrA in blue, the xenon atom in red, water in white, and lipids in gray.

**Figure 5 molecules-28-01424-f005:**
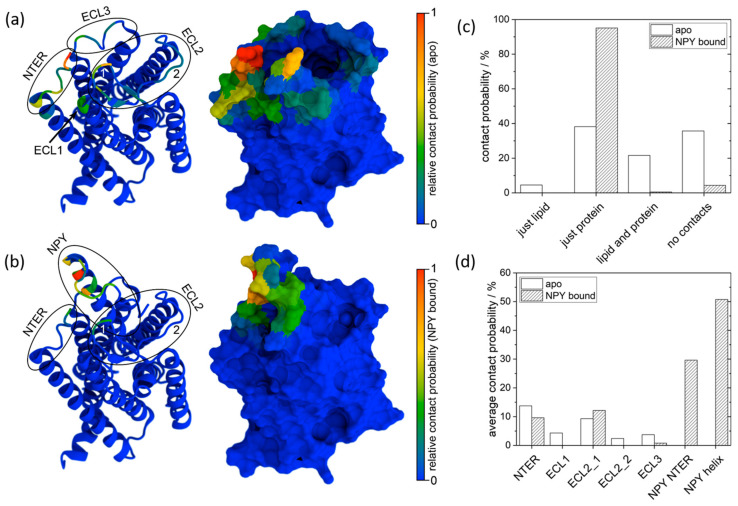
Projection of the number of contact instances onto the protein surface of apo Y2R-A202C-CrA (**a**) and the Y2R-A202C-CrA/ NPY complex (**b**) in cartoon (left) and surface (right) representations according to the indicated color scales. Note that the color scales in (**a**,**b**) were both normalized individually and are thus not directly comparable. The probability of contact between xenon and all system components is shown in (**c**) for apo Y2R-A202C-CrA (white) and the Y2R-A202C-CrA/ NPY complex (striped). The probabilities of contact with the different receptor sites observed in (**a**,**b**) are quantified in (**d**).

**Figure 6 molecules-28-01424-f006:**
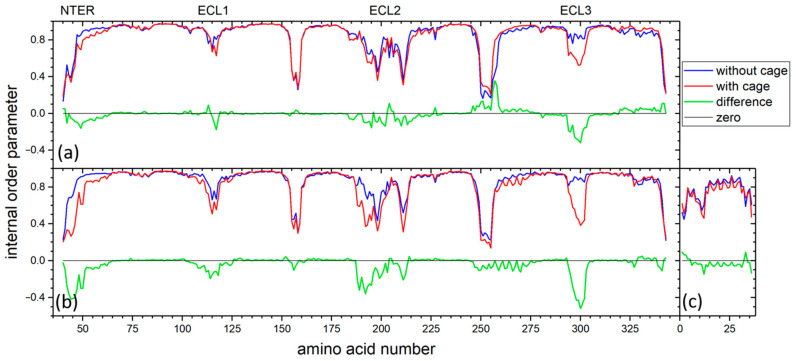
Internal order parameters of all amino acids in apo Y2R-A202C (**a**) and for both Y2R-A202C (**b**) and NPY (**c**) in the Y2R-A202C/NPY complex. Data in presence of construct are shown in red, while data without CrA-Cy3-mal are shown in blue. Their difference is shown in green, while a line at zero is shown in black for reference.

**Table 1 molecules-28-01424-t001:** Parameters fitted to the z-spectra of conjugate alone and in complex with NPY as superposition of 5 and 6 individual resonances, respectively. For comparison, the parameters of hpXe bound to the free CrA-Cy3-mal construct incubated in a buffer solution of bicelles and those of hpXe bound to free CrA-COOH incubated in a buffer solution of Y2R embedded in bicelles are listed.

Sample	Count	Position *b*/ppm	Amplitude *B*	Width *a/π*/Hz
conjugate	1	103.3 ± 2.0	0.069 ± 0.047	820 ± 916
conjugate, NPY inc.	1	104.9 ± 0.4	0.089 ± 0.027	322 ± 210
conjugate	2	91.9 ± 2.1	0.077 ± 0.031	947 ± 797
conjugate, NPY inc.	2	92.4 ± 0.5	0.136 ± 0.018	704 ± 159
conjugate	3	77.5 ± 0.1	2.300 ± 0.140	573 ± 42
conjugate, NPY inc.	3	77.5 ± 0.1	2.120 ± 0.075	618 ± 33
conjugate	4	67.2 ± 0.8	0.163 ± 0.131	242 ± 362
conjugate, NPY inc.	4	70.0 ± 0.8	0.181 ± 0.050	466 ± 267
conjugate	5	58.8 ± 3.4	0.092 ± 0.054	1570 ± 1306
conjugate, NPY inc.	5	63.0 ± 0.7	0.182 ± 0.027	785 ± 179
conjugate, NPY inc.	6	49.5 ± 0.4	0.109 ± 0.018	417 ± 156
bicelles, CrA-Cy3-mal inc.		77.9 ± 0.1	2.206 ± 0.100	497 ± 21
Y2R-bicelles, CrA-COOH inc.		77.3 ± 0.5	-	320 ± 20
Y2R-bicelles, CrA-COOH inc.		63.7 ± 0.5	-	24 ± 2

**Table 2 molecules-28-01424-t002:** List of amino acids in Y2R-A202C-CrA and in NPY subjected to contact at each detected contact site.

Contact Site	Apo Y2R-A202C-CrA	Y2R-A202C-CrA/NPY
NTER	L40, I41, K45	L40, I41, S43
ECL1	L112, M113, G114	
ECL2_1	I195, P196, F198	I195, P196, F198
ECL2_2	W207, P208, G209	
ECL3	Q296, D299, L300	L300, K301, E302
NPY NTER		G9, E10, A12
NPY helix		A14, M17, Y21

## Data Availability

The data presented in this study are available on request from the corresponding author.

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
