# Peer review of "Towards Probing Conformational States of Y2 Receptor Using Hyperpolarized 129Xe NMR"

_molecules, 2023, doi:10.3390/molecules28031424_

Round 1
Reviewer 1 Report
The Manuscript can be accepted after addressing some concerns
1. Abstract is not much clear on the results section.
2. How did the authors selected the protein the protein structure ? Did they try with other PDB codes for the protein?
3. How the complex prepared for MD?
4. How the authors find and used the cut off distance in Simulation .
5. What is box size and why they have used that size.
6. how the SASA and Radius of Gyration can effect and be important in such studies ?
7. Why authors used the 298 K Temperature instead of using 310 k
Reviewer 2 Report
The manuscript does not contain Figure 2, which is a major problem. Without this figure a large part of the manuscript could not be reviewed. Next to this major problem a couple of minor problems were found related to the figures. On Figure 1B the lower part was truncated, so a part of the sequence, including the site of an original cysteine, is not shown. The font used on the axis of Figure 3 are too large. The choice of black and gray color is not a good idea. In Figure 5 C and D the axis text in not readable, a different color choice would here be also welcome.
Please give more details on the Y2R-Cys-dpl construct. Almost every cysteine residues were mutated to alanines, except C103. Most of the cytein side chains are pointing toward the hydrophobic interior of the membrane, so this choice is well-founded. In the case of C151 the mutated residue is missing from Figure 1B. This residue is probably already at the edge of the lipid bilayer, so the mutated residue has not to be hydrophobic, but the choice of residue is unknown. Cysteine 103, which points not into the lipid bilayer in the proximity of the NPY ligand, was mutated to a serine.
The alanine 202 residue was mutated to cysteine, serving as an attachment point for cryptophane-A (CrA). This is the direct neighbor of the disulphide forming cysteine 203 residue. Since this region of the protein is disordered in the absence of the NPY ligand, is it not possible that the introduced C202 can form disulphide bridges with the cysteine 123 residue, introducing a conformational strain into the structure? Were any validations done if the disulphide bond is formed between the appropriate residues? The more distant Ile 200 residue might have been a better choice as point of CrA attachment.
Since the attached CrA cage is bulky it could significantly modify the structural dynamics of the Y2R-NPY complex. Were any MD simulations performed on the Y2R-NPY complex in the absence of the CrA cage? Simulations or at least a comparison of the experimental Y2R-NPY complex and representative structures from the MD simulations would be needed to strengthen the usability of the Y2R-A202C-CrA construct as a model system to study the structural dynamics of the native Y2R-NPY complex.
To help the interpretation of the (missing) spectra, several independent relatively long (500 ns) MD simulations were performed, where the first 100 ns (equilibration) were not analyzed. Please include some representative (Y2R or NPY binding site RMSD vs time) data to support this protocol. I would advise to attach this and possibly some other data as supplement to the manuscript. In paragraph 2.4 less speculations about NPY contacts are given. Would increasing the number of contacting residues not change this observation? In this paragraph it is speculated that not all receptors are occupied by NPY. Could a higher than 2x molar excess concentration of NPY be used to decrease this effect?
Different contact sites were characterized using the 3 most frequently “touched” residues. For NPY as a single contact site, 3 out of the 37 residues seem to be not enough. I understand that increasing the number of residues would make assignation ore complicated. Could a residence time/contact probability threshold be used to define the “contact residues” in such a way, that their number might be different for distinct contact sites? Please include all residues identified within all contact sites in the supplement (possibly with a quantitative descriptor), to be able to see if the 3 residue reduction is valid.
Please explain why 8.5 Å distance definition was used for contact definition. Being xenon inside the CrA cage, this distance seem to be quite low. Does it have any spectroscopy reason? Were hydrogen or only heavy atom distances used to define contact?
Some final remarks:
How was the 36th (TYC) residue of NPY handled in the MD simulations?
Please include the “boundary” distance used to create the simulation box.
Please include a reference about the unstructured nature of the ECL2 loop.
Please explain why linear approximation is not valid for resonance 3 with the largest amplitude.
Please correct some missing and extra spaces in the text.
Round 2
Reviewer 1 Report
It can be published
Reviewer 2 Report
Thank you for the additional work, which was put into the revised version!